# miR-182-5p promotes hepatocyte-stellate cell crosstalk to facilitate liver regeneration

Ting Xiao[1,5], Wen Meng [1,5 ✉], Zhangliu Jin[1,2], Jing Wang[1], Jiangming Deng[1], Jie Wen[1], Bilian Liu[1], Meilian Liu [1,3], Juli Bai[1,4] & Feng Liu [1 ✉]

A unique feature of the liver is its high regenerative capacity, which is essential to maintain liver homeostasis. However, key regulators of liver regeneration (LR) remain ill-defined. Here, we identify hepatic miR-182-5p as a key regulator of LR. Suppressing miR-182-5p, whose expression is significantly induced in the liver of mice post two-thirds partial hepatectomy (PH), abrogates PH-induced LR in mice. In contrast, liver-specific overexpression of miR-182-5p promotes LR in mice with PH. Overexpression of miR-182-5p failed to promote proliferation in hepatocytes, but stimulates proliferation when hepatocytes are cocultured with stellate cells. Mechanistically, miR-182-5p stimulates Cyp7a1-mediated cholic acid production in hepatocytes, which promotes hedgehog (Hh) ligand production in stellate cells, leading to the activation of Hh signaling in hepatocytes and consequent cell proliferation. Collectively, our study identified miR-182-5p as a critical regulator of LR and uncovers a Cyp7a1/cholic acid-dependent mechanism by which hepatocytes crosstalk to stellate cells to facilitate LR.

[1] National Clinical Research Center for Metabolic Diseases, Metabolic Syndrome Research Center, Key Laboratory of Diabetes Immunology, Ministry of Education, and Department of Metabolism and Endocrinology, The Second Xiangya Hospital of Central South University, Changsha 410011 Hunan, China. [2] Department of Biliopancreatic Surgery and Bariatric Surgery, The Second Xiangya Hospital of Central South University, Changsha 410011 Hunan, China. [3] Department of Biochemistry and Molecular Biology, University of New Mexico Health Sciences Center, Albuquerque, NM 87131, USA. [4] Department of Pharmacology, University of Texas Health Science Center at San Antonio, San Antonio, TX 78229, USA. [5]These authors contributed equally: Ting Xiao, Wen Meng. ✉email: 122501006@csu.edu.cn; liuf001@csu.edu.cn

L iver is a unique organ that can regenerate its mass in response to injury[1–4]. Liver regeneration (LR) is an orchestrated process consisting of three major phases, initiation, proliferation, and termination. In the initiation phase, quiescent hepatocytes are primed to achieve competence for proliferation. In the proliferation phase, the hepatocyte population is expanded until it senses the required cell mass, leading to the termination of the proliferation process. The different phases during LR are mainly regulated by various cytokine-, growth factor-, and hepatic hormone-induced signaling pathways[2,4,5]. However, how these signaling pathways are initiated and coordinately regulated during LR remains to be further elucidated.

Being a heterogeneous tissue, the liver is composed of multiple cell types such as Kupffer cells (resident macrophages), hepatic stellate cells (HSCs), hepatic sinusoidal endothelial cells, and recruited immune cells, in addition to hepatocytes[6]. The microenvironment of the liver, which is maintained by dynamic crosstalk among these different cell types, has a profound impact on LR. One of the key players in regulating LR is HSCs, which have been shown to promote hepatocyte proliferation by secreting a variety of growth factors and cytokines after partial hepatectomy (PH)[7–9]. However, the precise mechanism by which PH promotes growth factor and cytokine production in HSCs remains largely unclear.

As a pleiotropic miRNA, miR-182-5p is highly conserved across species and plays a critical role in regulating various biological processes such as immune response, DNA repair, and cancer development[10–12]. Our recent study found that miR-182-5p in adipose tissue promotes beige fat thermogenesis via crosstalk between adipocytes and macrophages[13]. New evidence has emphasized a link between miR-182-5p and liver disease, such as hepatic steatosis, alcoholic hepatitis, and hepatocellular carcinoma (HCC)[14–16]. miR-182-5p has been shown to increase cell growth and proliferation in several carcinoma cells[17–19]. A recent high-throughput sequencing study showed that the expression of miR-182-5p is greatly induced in rat liver after PH[20]. However, whether and how miR-182-5p plays a role in LR is unknown.

In this study, we identified miR-182-5p as a key regulator of LR in mice. In addition, we demonstrate that miR-182-5p facilitates LR by promoting the hepatocyte-HSC crosstalk via stimulating the cholic acid (CA)-India/hedgehog signaling axis.

## Results

### miR-182-5p deficiency suppresses mouse hepatocyte proliferation after 2/3 PH. Based on the finding that the expression of miR-182-5p is greatly induced in rat liver after PH[20], we asked whether miR-182-5p plays a role in PH-induced LR in mice. Given that homozygous miR-182-5p knockout mice are defective in retinal function[21], we performed PH on miR-182-5p heterozygous knockout mice (miR-182-5p$^{KO}$), which showed an ~50% reduction in miR-182-5p expression levels (Supplementary Fig. 1a) but did not show any detectable abnormalities in growth, morphology, and metabolism compared to their control (WT) mice under normal growth conditions[13]. As shown in Fig. 1a, the ratio of liver to body weight of the miR-182-5p$^{KO}$ mice was greatly reduced compared to that of the WT control mice on day 7 after PH. Consistently, the proliferating Ki67+ hepatic cell numbers were significantly decreased in miR-182-5p$^{KO}$ mice compared to the WT control mice at day 3 after PH (Fig. 1b), an important phase of cell proliferation[1,4]. To further confirm the role of miR-182-5p in hepatocyte proliferation after PH, we examined the expression of several cyclin genes in the liver of miR-182-5p$^{KO}$ mice and their control mice. The mRNA levels of Cyclin D1 (CcnD1), which drive hepatocyte proliferation following PH[2], Cyclin A2 (CcnA2), B1 (CcnB1), and E1 (CcnE1) were

significantly lower in the liver of miR-182-5p$^{KO}$ mice compared to their control mice at day 3 after PH (Fig. 1c). Consistent with this finding, miR-182-5p deficiency greatly reduced PH-induced expression and/or phosphorylation of ERK, S6, 4EBP1 and Cyclin D1 in the liver of miR-182-5p$^{KO}$ mice compared to their WT control mice (Fig. 1d), indicating that miR-182-5p deficiency suppresses hepatocyte proliferation after 2/3 PH in mice. No significant difference in serum alanine aminotransferase and aspartate transaminase levels was observed between miR-182-5p$^{KO}$ mice and their control mice (Fig. 1e), suggesting that miR-182-5p deficiency had no major adverse effect on liver function. Taken together, these results reveal that miR-182-5p plays a key role in PH-induced hepatocyte proliferation.

### Liver-specific overexpression of miR-182-5p promotes LR in mice. To characterize the cell types in the liver in which miR-182-5p expression is induced by PH, we examined the expression of miR-182-5p in mouse hepatocytes and non-parenchymal cells (NPC) during LR. We found that PH greatly induced the expression of miR-182-5p in mouse hepatocytes isolated from WT mice, which peaked at ~3 days post PH (Supplementary Fig. 1b). Little miR-182-5p expression was observed in NPCs isolated from WT mice after PH (Supplementary Fig. 1b), demonstrating a critical role of hepatocellular miR-182-5p in LR. To further test this, we generated transgenic mice in which miR-182-5p is specifically overexpressed in hepatocytes (miR-182-5p$^{LTG}$) by crossing miR-182-5p-Tg$^{flox/flox}$ mice with albumin-cre mice. miR-182-5p level was significantly increased in the liver but not in other tissues of the miR-182-5p$^{LTG}$ (TG) mice (Fig. 2a). Hepatic-specific overexpression of miR-182-5p significantly increased the mRNA levels of Cyclin D1, B1, and E1 in the liver of mice at day 3 following PH (Fig. 2b). miR-182-5p overexpression also significantly increased the proliferating Ki67+ hepatic cell numbers (Fig. 2c) and mRNA levels (Fig. 2d), as well as the phosphorylation of ERK, S6, 4EBP1, and the expression of Cyclin D1 (Fig. 2e) in the liver of the PH-treated mice. Consistent with these findings, the liver-to-body weight ratio of the miR-182-5p$^{LTG}$ mice was greatly increased compared to control (WT) mice at day 7 after PH (Fig. 2f). These results further demonstrate that hepatocyte miR-182-5p plays a critical role in promoting LR in vivo.

### miR-182-5p promotes hepatocyte proliferation via an indirect mechanism. To determine the cellular mechanism by which hepatocyte miR-182-5p facilitates LR, we transfected miR-182-5p mimic or its negative control in mouse primary hepatocytes treated with epidermal growth factor (EGF), which is necessary to induce hepatocyte proliferation in vitro[22]. To our surprise, overexpression of the miR-182-5p mimic had no significant effect on cyclin gene expression in primary hepatocytes (Supplementary Fig. 2a, b). In addition, the EdU-positive nuclei were comparable between miR-182-5p mimic and control mimic-treated cells (Supplementary Fig. 2c). Furthermore, inhibition of miR-182-5p by a miR-182-5p inhibitor (Supplementary Fig. 2d), a synthetic double-stranded miRNA that inhibits endogenous miR-182-5p, had no significant effect on the expression levels of cyclin genes and the numbers of EdU-positive cells in EGF-treated primary hepatocytes (Supplementary Fig. 2e, f). These results suggest that the in vivo effect of miR-182-5p on hepatocyte proliferation after PH may be mediated by an indirect mechanism involving the crosstalk between hepatocytes and neighboring cells in the liver. To test this possibility, we co-cultured mouse non-parenchymal cells (NPCs) with primary hepatocytes isolated from WT and miR-182-5p$^{KO}$ mice (Fig. 3a). We found that EGF-induced cyclin gene expression is significantly suppressed in miR-182-5p-deficient primary

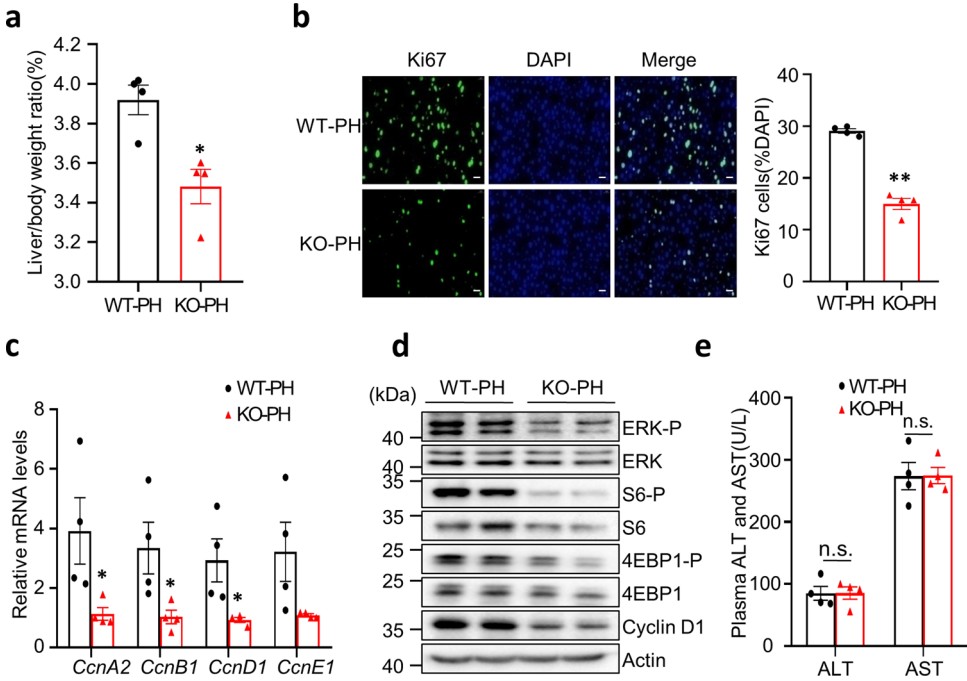

**Fig. 1 miR-182-5p deficiency suppresses mouse hepatocyte proliferation after 2/3 PH.** Male miR-182-5p[KO] (KO) and WT mice (8-week old) were subjected to 2/3 PH. **a** Liver to body weight ratio of miR-182-5p KO and WT mice (7d after PH; n = 4/group). **b** Liver sections were immunostained with an anti-Ki67 antibody. Ki67[+] positive cells were counted and normalized to total DAPI[+] cells (3d after PH; n = 4/group; scale bar: 20 μm). **c** qRT-PCR analyses of *cyclin* gene expression in the liver of miR-182-5p KO and WT mice (3d after PH; n = 4/group). **d** Lysates of liver homogenates were examined for proliferative signaling by western blot analysis (3d after PH). **e** Serum AST and ALT levels were measured in miR-182-5p KO and WT mice (3d after PH; n = 4/group). Error bars in all experiments represent SEM; Significance was determined by unpaired two-tailed Student's *t* test. *P < 0.05, **P < 0.01; n.s.: not significant.

hepatocytes (Hep[182KO]) compared to WT primary hepatocytes (Hep[WT]) co-cultured with NPCs (Fig. 3b, c). In addition, over-expression of miR-182-5p in hepatocytes greatly increased cyclin gene expression when the cells were co-cultured with NPCs (Fig. 3d, e). To further validate the hepatocyte role of miR-182-5p in cell proliferation, we co-cultured mouse primary hepatocytes with NPCs isolated from WT and miR-182-5p KO mice (Fig. 3f). Knocking down miR-182-5p in NPCs had no significant effect on EGF-stimulated cyclin gene expression in mouse primary hepatocytes (Fig. 3g, h), further confirming the role of hepatocyte miR-182-5p in regulating the crosstalk between hepatocytes and NPCs to promote LR.

**miR-182-5p promotes hepatocyte proliferation via a stellate cell-dependent mechanism.** The liver is composed of several major cell types including hepatocytes, resident macrophages (Kupffer cells), hepatic stellate cells (HSCs), and liver sinusoidal endothelial cells[6]. Both Kupffer cells and stellate cells have been found to contribute to PH-induced hepatocyte proliferation in mice[4,23–25]. To determine whether Kupffer cells are involved in miR-182-5p-mediated hepatocyte proliferation after PH, we examined cyclin gene expression in primary hepatocytes isolated from wild-type (Hep[WT]) or miR-182-5p[+/−] mice (Hep[182KO]) co-cultured with bone marrow-directed macrophages (BMDMs). No significant difference in the expression of cyclin genes and the proliferation signature gene *Mki67* was observed between wild-type and miR-182-5p-suppressed primary hepatocytes co-cultured with BMDMs (Supplementary Fig. 3a, b). On the other hand, miR-182-5p deficiency in primary hepatocytes significantly reduced the expression of cyclin genes and *Mki67* in the hepatocytes co-cultured with primary HSCs (Fig. 4a, b) or the HSC cell line LX2 (Supplementary Fig. 3c, d). Conversely, over-expression of miR-182-5p in primary hepatocytes co-cultured

with primary HSCs (Fig. 4c–e) or LX2 cells (Supplementary Fig. 3e–g) greatly induced HSC activation as demonstrated by increased expression of α-SMA (*Acta2*) and *Col1a1*, which encode α-smooth muscle actin and collagen type I alpha 1, respectively, in primary HSCs or LX2 cells (Fig. 4e and Supplementary Fig. 3 g), concurrently with increased mRNA levels of *cyclin* (Fig. 4c and Supplementary Fig. 3e) and *Mki67* (Fig. 4d and Supplementary Fig. 3f) in primary hepatocytes. Collectively, these findings suggest that HSCs were required for PH-induced and miR-182-5p-mediated hepatocyte proliferation in vivo.

**miR-182-5p promotes hepatocyte proliferation by stellate cell-dependent activation of hedgehog signaling.** To further dissect the mechanism by which miR-182-5p promotes hepatocyte proliferation, we co-cultured HSCs (LX2) with mouse primary hepatocytes (LX2 + Hep[WT]) or with hepatocytes isolated from the liver-specific miR-182-5p overexpression transgenic mice (LX2 + Hep[182/LTG]). We then collected the conditioned media (CM) of the co-cultured cells and used them to treat mouse primary hepatocytes (Fig. 5a). The CM of LX2 + Hep[182/LTG] cells had a much greater promoting effect on hepatic Cyclin D1 protein expression (Fig. 5b) and the EdU-positive cell numbers (Fig. 5c) compared to the CM of LX2 + Hep[WT] cells. These findings suggest the presence of a secretory factor(s) in the CM of HSCs which is/are induced by the co-cultured hepatocytes overexpressing miR-182-5p. The stimulatory effect of the CM from LX2 + Hep[182/LTG] on hepatocyte proliferation could be blocked by heat-denaturation treatment (Supplementary Fig. 4a, b), suggesting that a secretory protein molecule(s) in the stellate CM may be involved in promoting mouse hepatocyte proliferation.

HGF and TGFβ1[26] have been shown to play critical roles in LR after 2/3 PH[4,27]. However, we found that miR-182-5p deficiency in mice had no significant effect on the mRNA levels of HGF

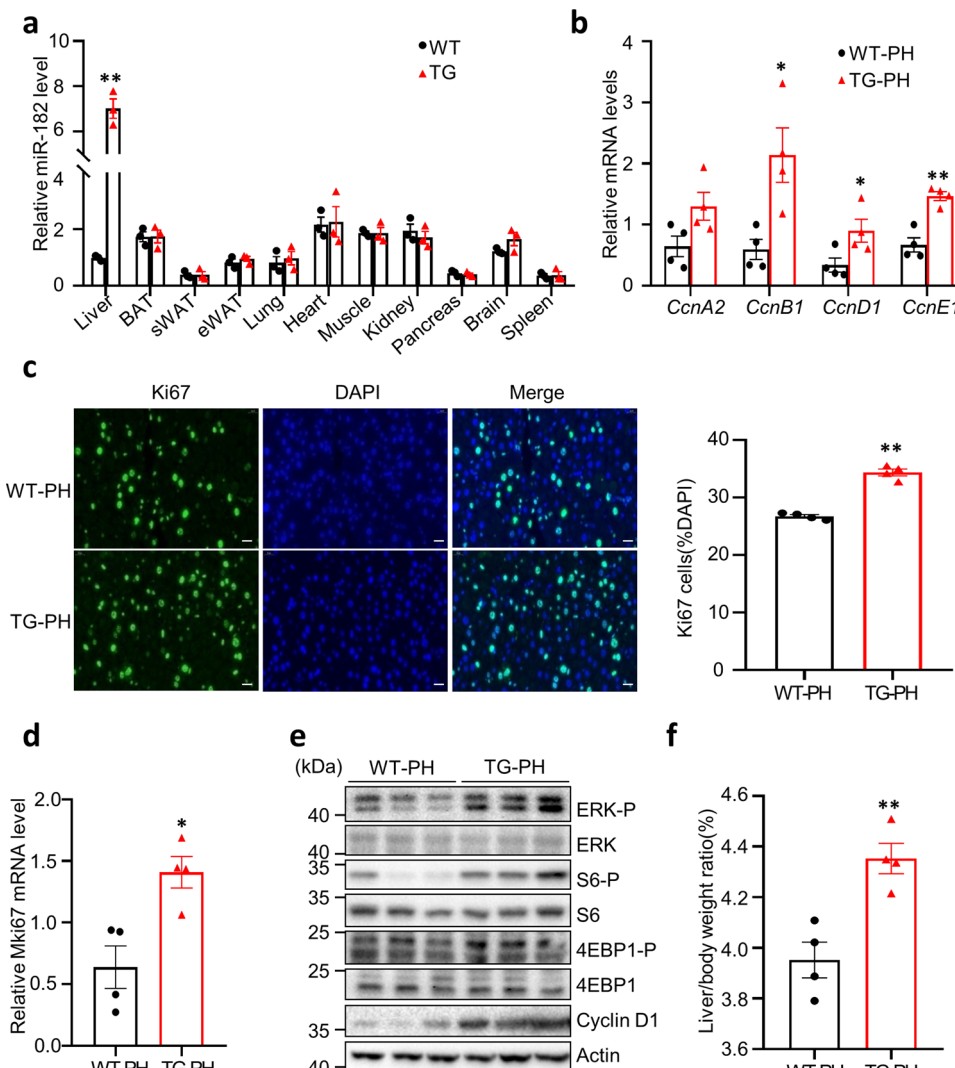

**Fig. 2 Liver-specific overexpression of miR-182-5p promotes hepatocyte proliferation in mice. a** qRT-PCR analyses of miR-182-5p level in liver and other tissues of liver-specific miR-182-5p overexpression (TG) and WT male mice ($n = 3$/group). Liver-specific miR-182-5p overexpression (TG) and WT male mice (8-week old) were subjected to 2/3 PH. **b** qRT-PCR analyses of *cyclin* genes expression in the liver of TG and WT mice (3d after PH; $n = 4$/group). **c** Liver sections were immunostained with an anti-Ki67 antibody and Ki67[+] cells were counted and normalized to total DAPI[+] cells (3d after PH; $n = 4$/group; scale bar: 20 μm). **d** qRT-PCR analyses of *Mki67* gene expression in the liver of TG and WT mice (3d after PH; $n = 4$/group). **e** Western Blot analysis of proliferative signaling in the liver of TG and WT mice (3d after PH). **f** Liver to bodyweight ratio of TG and WT mice (7d after PH; $n = 4$/group). Error bars in all experiments represent SEM; Significance was determined by unpaired 2-tailed Student's $t$ test. *$P < 0.05$, **$P < 0.01$.

(Supplementary Fig. 4c) and TGFβ (Supplementary Fig. 4d). On the other hand, the mRNA levels of hedgehog (Hh) ligands such as Indian (Ihh) and to a lesser extent the Sonic Hh (Shh) were significantly suppressed in the liver of miR-182-5p[KO] mice compared to WT control mice at day 3 after 2/3 PH (Fig. 5d and Supplementary Fig. 4e). Given that both Ihh and Shh are expressed in activated HSCs[4,28], we postulated that miR-182-5p may promote LR by increasing Ihh expression and secretion in HSCs, which subsequently induces Hh signaling in hepatocytes. Indeed, the expression level of Ihh was significantly upregulated in LX2 co-cultured with miR-182-5p-overexpressed hepatocytes compared to LX2 co-cultured with WT hepatocytes (Fig. 5e). To confirm the role of hedgehog signaling in miR-182-5p-induced hepatocyte proliferation, we treated hepatocytes with CMs of LX2 activated by co-culturing with Hep[182/LTG] cells. The promoting effect of the LX2 CM on hepatocyte proliferation was significantly suppressed by either pharmacological inhibition of hedgehog signaling with cyclopamine (5 μM) for 24 h (Fig. 5f, g), or by

siRNA-mediated suppression of the Hh receptor Smo in primary hepatocytes (Supplementary Fig. 4f and Fig. 5h, i). Taken together, these results demonstrate that hepatic miR-182-5p promotes mouse hepatocyte proliferation via HSC-dependent activation of the Hh signaling pathway.

**miR-182-5p promotes hepatocyte proliferation by enhancing CA-mediated activation of HSCs.** To determine the mechanism by which hepatic miR-182-5p activates HSCs in the liver, we performed RNA-seq analysis in the liver of miR-182-5p[KO] mice and wild-type control mice at day 3 after 2/3 PH. Volcano plot analysis of the RNA-seq data with a twofold change identified 2162 differentially expressed genes (DEGs), among them 545 were upregulated, and 1617 were downregulated (Supplementary Fig. 5a). To functionally annotate the DEGs, we aligned all DEGs against the Kyoto Encyclopedia of Genes and Genomes (KEGG) database. KEGG pathway enrichment analysis identified several pathways that were greatly enriched in miR-182-5p-deficient

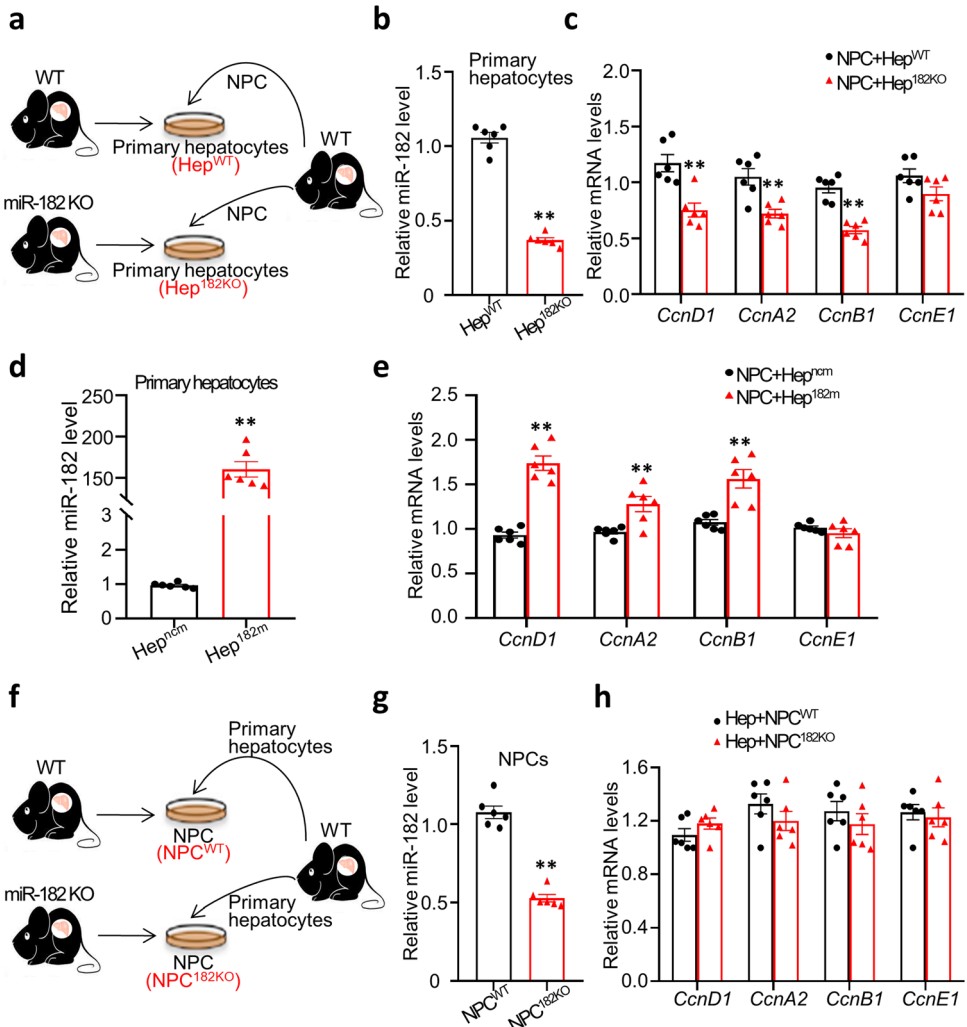

**Fig. 3 miR-182-5p promotes hepatocyte proliferation via an indirect mechanism involving non-parenchymal cells. a** A schematic diagram of primary hepatocytes from miR-182-5p KO and WT mice co-cultured with NPCs of C57BL/6 J mice in the presence of EGF. **b** The expression of miR-182-5p in primary hepatocytes from miR-182-5p KO (Hep$^{182KO}$) and WT mice (Hep$^{WT}$) co-cultured with NPCs of C57BL/6 J mice ($n = 6$/group). **c** qRT-PCR analyses of *cyclin* genes expression in primary hepatocytes from miR-182-5p KO mice and their control mice co-cultured with NPCs of C57BL/6 J mice ($n = 6$/group). miR-182-5p mimic (182 m) or its negative control (ncm) were overexpressed in primary hepatocytes isolated from C57BL/6 J mice, and following co-cultured with NPCs of C57BL/6 J mice. **d** The expression of miR-182-5p in primary hepatocytes ($n = 6$/group). **e** qRT-PCR analyses of *cyclin* genes expression in miR-182-5p-overexpressed (Hep$^{182m}$) and their control (Hep$^{ncm}$) primary hepatocytes co-cultured with NPCs of C57BL/6 J mice ($n = 6$/group). **f** A schematic diagram of NPCs from miR-182-5p KO and WT mice co-cultured with primary hepatocytes of C57BL/6 J mice in the presence of EGF. **g** The expression of miR-182-5p in NPCs from miR-182-5p KO (NPC$^{182KO}$) and WT mice (NPC$^{WT}$) co-cultured with primary hepatocytes ($n = 6$/group). **h** qRT-PCR analyses of *cyclin* gene expression in primary hepatocytes co-cultured with NPCs from miR-182-5p KO mice and their control mice ($n = 6$/group). Error bars in all experiments represent SEM; Significance was determined by unpaired two-tailed Student's *t* test. **$P < 0.01$.

mouse liver, and the most enriched pathway is related to primary bile acid (BA) biosynthesis (Supplementary Fig. 5b). In agreement with these findings, the serum concentration of BA was greatly decreased in miR-182-5p$^{KO}$ mice at day 3 after 2/3 PH compared to wild-type control mice (Fig. 6a). In contrast, BA levels in serum (Fig. 6b) or liver (Fig. 6c) were significantly increased in the miR-182-5p$^{LTG}$ mice at day 3 after 2/3 PH. To determine the potential role of BAs in miR-182-5p-induced activation of HSCs in the liver, we performed high-throughput profiling of BAs in the liver of the miR-182-5p$^{LTG}$ mice and their control mice (WT) on day 3 after 2/3 PH, using liquid chromatography/mass spectrometry (LC/MS). Heatmap (Fig. 6d) and volcano plot (Fig. 6e) analysis revealed that three primary BAs, CA, glycocholic acid (GCA) and taurocholic acid (TCA), were significantly upregulated in the liver of miR-182-5p TG mice compared to the wild-type control mice

at day 3 after 2/3 PH (Fig. 6f and Supplementary Fig. 5c, d). Treating primary HSCs (Fig. 6g, h) or LX2 (Supplementary Fig. 5e, f) with CA for 48 h, but not GCA (Supplementary Fig. 5g, h) or TCA (Supplementary Fig. 5i, j), markedly induced the expression of *α-SMA* and *Ihh*, demonstrating a specific role of CA in mediating hepatic miR-182-5p-induced HSC activation.

To determine the signaling pathway(s) involved in CA-mediated HSC activation, we examined the effect of CA on the expression of several CA receptors in stellate cells, including FXR, TGR5, S1PR2, and VDR. Consistent with previous reports[29,30], we found that the expression levels of FXR, S1PR2, and VDR were low in stellate cells (Supplementary Fig. 5k) and CA treatment had no significant effect on the expression of these molecules (Supplementary Fig. 5l). However, CA treatment greatly increased the expression of TGR5 in stellate cells

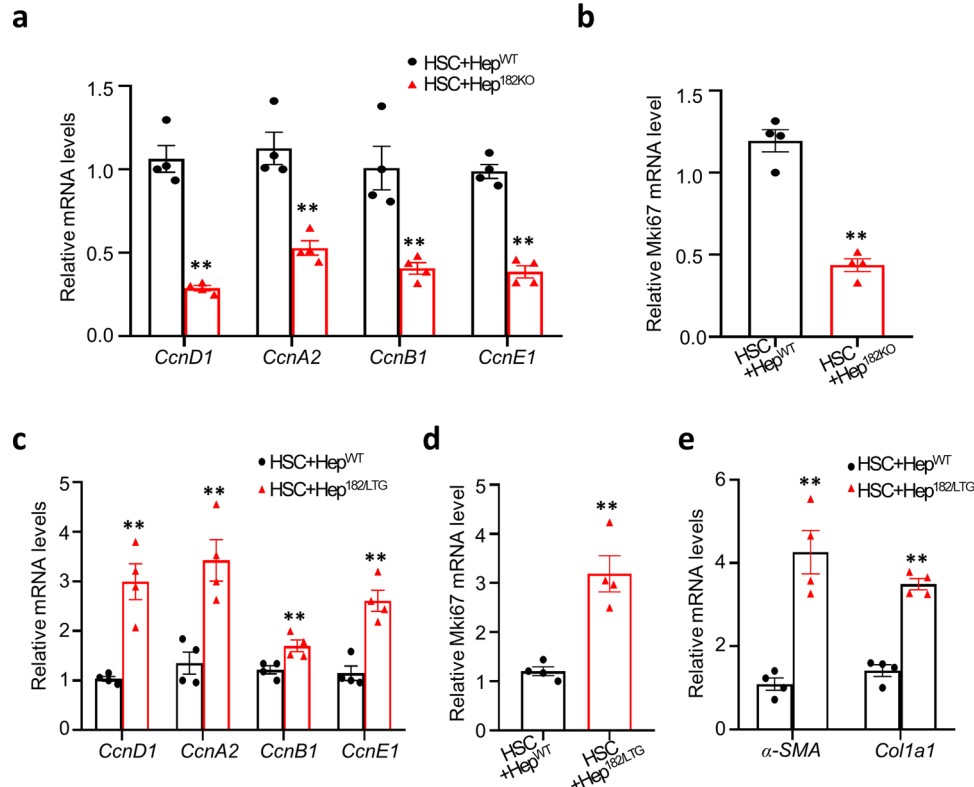

**Fig. 4 miR-182-5p promotes hepatocyte proliferation via a stellate cell-dependent mechanism.** qRT-PCR analyses of *cyclin* genes **a** and Mki67 **b** expression in primary hepatocytes from miR-182-5p[KO] mice (Hep[182KO]) and their control mice (Hep[WT]) co-cultured with primary HSCs in the presence of EGF (*n* = 6/group). Primary hepatocytes from miR-182-5p[LTG] mice (Hep[182/LTG]) and their control mice (Hep[WT]) co-cultured with primary HSCs. qRT-PCR analyses of *cyclin* genes **c** and Mki67 gene **d** expression in primary hepatocytes. **e** qRT-PCR analyses of HSC activation marker genes expression in primary HSCs (*n* = 4/group). Error bars in all experiments represent SEM; significance was determined by unpaired 2-tailed Student's *t* test. \*\**P* < 0.01.

(Supplementary Fig. 5l). Suppressing TGR5 expression by siRNA in stellate cells (Fig. 6i) greatly reduced CA-induced expression of *α-SMA* (Fig. 6j) and *Ihh* (Fig. 6k), revealing a promoting role of TGR5 in CA-induced HSC activation and hedgehog ligand production.

**Hepatic miR-182-5p targets Cyp7a1 to promote hepatocyte proliferation.** The rate of BA synthesis is mainly controlled by the classic and alternative BA synthesis pathways via the rate-limiting enzyme cholesterol 7α-hydroxylase (CYP7A1) and CYP27A1, respectively[31] (Supplementary Fig. 6a). RNA-seq analysis revealed that the expression levels of *Cyp7a1* and *Cyp27a1* genes were greatly decreased in the liver of miR-182-5p[KO] mice compared to wild-type control mice at day 3 after 2/3 PH (Fig. 7a), which was confirmed by real-time PCR (Fig. 7b). On the other hand, the mRNA levels of *Cyp7a1* but not *Cyp27a1* were significantly increased in the liver of TG mice compared with their control mice at day 3 after 2/3 PH (Fig. 7c). Consistently, overexpression of the miR-182-5p mimics significantly increased *Cyp7a1* but not *Cyp27a1* mRNA expression in hepatocytes (Fig. 7d, e). Moreover, the CYP7A1 protein level was markedly increased in miR-182-5p-overexpressed hepatocytes (Fig. 7f), suggesting that miR-182-5p exerts a cell-autonomous effect on *Cyp7a1* gene and protein expression. Interestingly, *Cyp7a1* expression was strongly suppressed at the early stages of LR but gradually increased at the proliferative phase, concurrently with increased miR-182-5p expression (Supplementary Fig. 6b). Heterozygous knockout of miR-182-5p had no significant effect on *Cyp7a1* expression at the early stages of LR but inhibited *Cyp7a1* expression at the proliferative phase during LR in mice

(Supplementary Fig. 6c), revealing a phase-specific role of miR-182-5p in regulating Cyp7a1 expression during LR.

miRNAs normally silence gene expression by promoting the degradation of mRNAs. Thus, we first examined whether miR-182-5p promotes *Cyp7a1* gene expression by suppressing negative regulators of *Cyp7a1*. Several negative regulators of the *Cyp7a1* gene have been reported, including *Fgfr4, Pparα, Nr0b2* (SHP), *Nr1h4* (FXR), and *Nr1l2* (PXR)[30,32]. However, overexpression of miR-182-5p had no significant effect on the expression of these negative regulators of *Cyp7a1* (Supplementary Fig. 6d). Based on the finding certain miRNAs could stimulate gene expression by targeting the promotors of these genes[33,34], we then asked whether miR-182-5p could directly stimulate the expression of the *Cyp7a1* gene. Bioinformatic analysis revealed that miR-182-5p could align to the promoter region of *Cyp7a1* gene (Fig. 7g). By luciferase reporter assay, we found that the miR-182-5p mimic increased the promoter activity of the *Cyp7a1* reporter gene but not the one with mutations in the seed sequence (Fig. 7h), revealing that miR-182-5p stimulated *Cyp7a1* gene expression by directly interacting with its promoter. To further characterize the role of Cyp7a1 in miR-182-5p-induced hepatocyte proliferation, we treated primary hepatocytes isolated from TG mice with Cyp7a1 siRNA (Supplementary Fig. 6e) and co-cultured these cells with primary HSCs. Suppressing *Cyp7a1* expression greatly reduced the expression levels of *α-SMA* (Fig. 7i) and *Ihh* (Fig. 7j) in primary HSCs as well as the promoting effect of HSC + Hep[182TG] on hepatocyte proliferation (Fig. 7k, l). Taken together, these results demonstrate that hepatocyte miR-182-5p-mediated Cyp7a1/CA signaling promotes hepatocyte proliferation by enhancing hedgehog signaling in stellate cells.

## Discussion

In the present study, we identified miR-182-5p as a critical regulator of LR. In addition, we demonstrate that the promoting effect of hepatic miR-182-5p on LR is mediated by a paracrine mechanism via cell-cell crosstalk in the liver. miR-182-5p expression is significantly stimulated by PH and overexpression of this molecule in hepatocytes enhances CA production, which triggers Hh ligand production from HSCs. Elevated Hh ligand levels

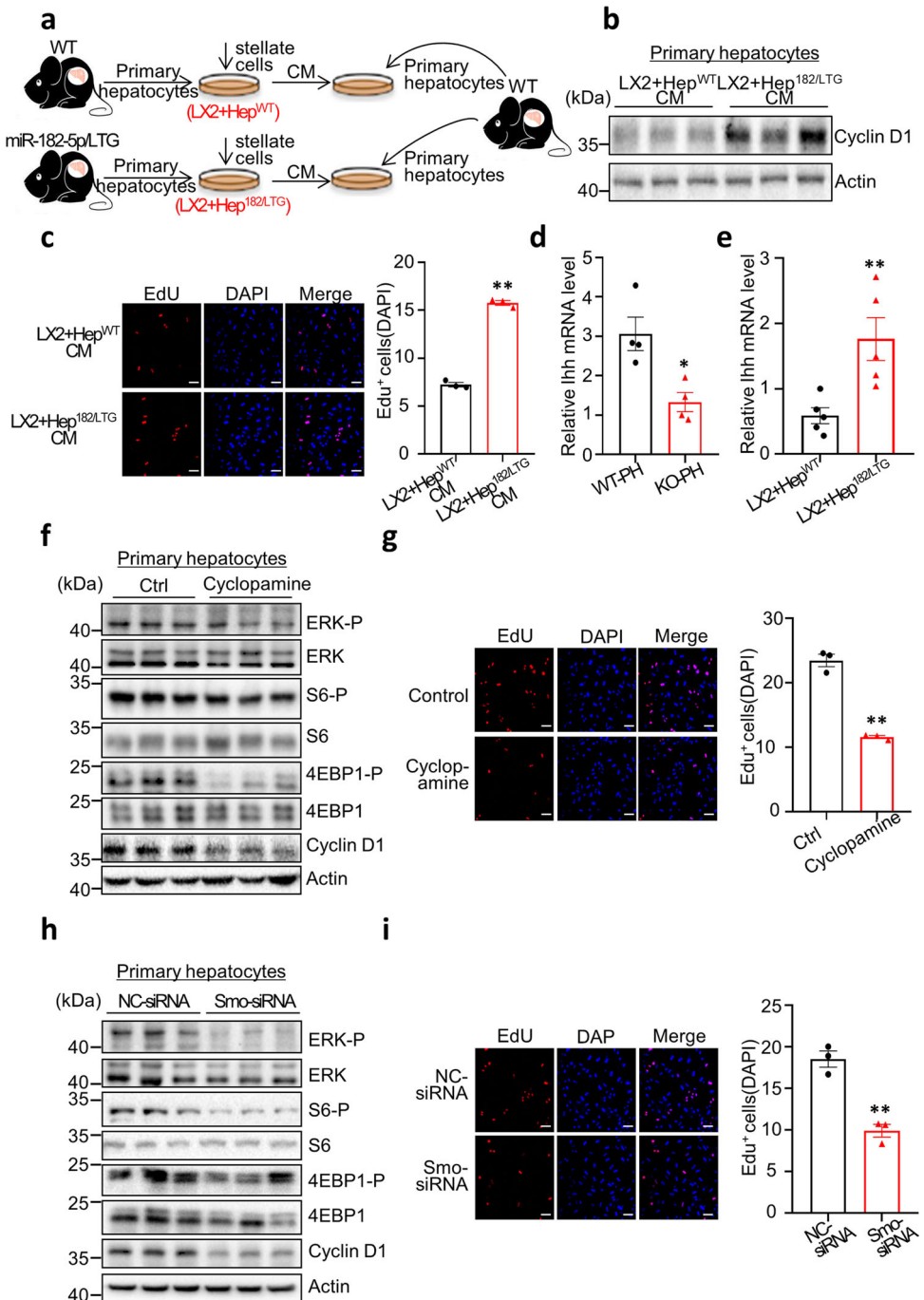

**Fig. 5 miR-182-5p promotes hepatocyte proliferation by stellate cell-dependent activation of hedgehog signaling. a** A schematic diagram of the co-culture experiment in which primary hepatocytes were treated with a conditioned medium (CM) of HSCs (LX2 cells) co-cultured with miR-182-5p-overexpressed hepatocyte (LX2 + Hep$^{182/LTG}$) or with that of their control (LX2 + Hep$^{WT}$) for 24 h. **b** Western blot analysis of Cyclin D1 level in primary hepatocytes. **c** EdU immunostaining of primary hepatocytes ($n = 3$/group; scale bar: 100 μm). **d** qRT-PCR analyses of *Ihh* gene expression in the liver of miR-182-5p KO and WT mice (3d after PH; $n = 4$/group). **e** qRT-PCR analyses of *Ihh* gene expression in LX2 co-cultured with primary hepatocytes from miR-182-5p TG mice (LX2 + Hep$^{182/LTG}$) or their control mice (LX2 + Hep$^{WT}$)($n = 4$/group). Primary hepatocytes were treated with a conditioned medium of stellate cells co-cultured with miR-182-5p-overexpressed hepatocyte treated with or without cyclopamine. **f** Western blot analysis of proliferative signaling in primary hepatocytes. **g** EdU immunostaining of primary hepatocytes ($n = 3$/group; scale bar: 100 μm). Smo siRNA or control siRNA (NC-siRNA)-treated primary hepatocytes were treated with a conditioned medium of stellate cells co-cultured with miR-182-5p-overexpressed hepatocyte. **h** Western blot analysis of proliferative signaling in primary hepatocytes. **i** EdU immunostaining of primary hepatocytes($n = 3$/group; scale bar: 100 μm). Error bars in all experiments represent SEM; Significance was determined by unpaired two-tailed Student's *t* test. *$P < 0.05$, **$P < 0.01$.

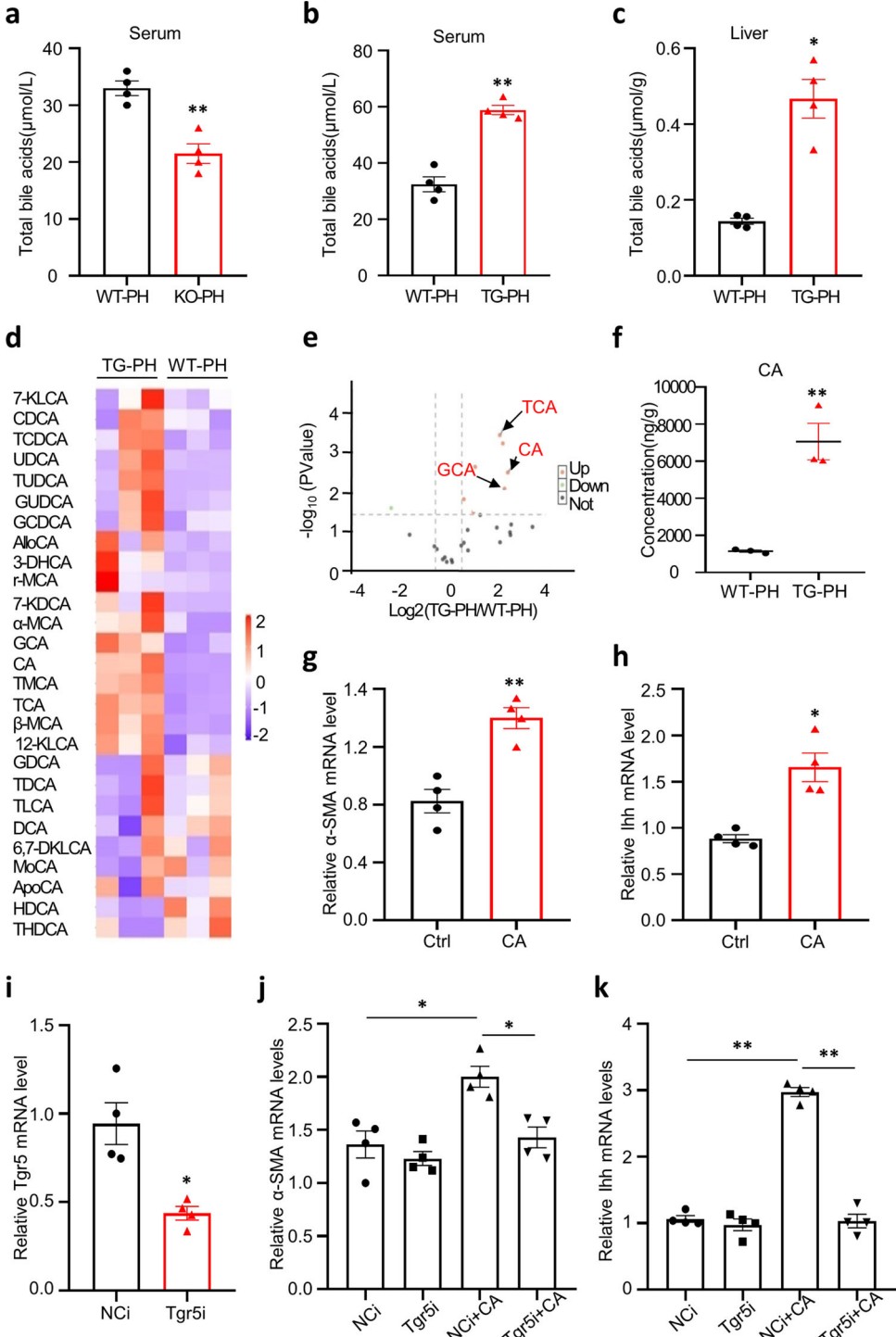

**Fig. 6 miR-182-5p promotes hepatocyte proliferation by enhancing cholic acid (CA)-mediated activation of HSCs. a** Total bile acid levels were measured in the serum of miR-182-5p KO and WT mice (3d after PH; *n* = 5/group). **b, c** Total bile acid levels were measured in the serum and liver of the miR-182-5p TG mice and their control mice(WT) at day 3 after 2/3 PH (*n* = 4/group). **d** Heatmap showing changes in the abundance of BAs composition in liver of TG and WT mice (3d after PH). Blue represents down-regulation while a red represents upregulation. **e** Volcano plot of the BAs composition. Red, significantly increased gene expression; green, significantly decreased; gray, no statistical differences. **f** LC/MS analyses of the CA level in liver of the miR-182-5p TG mice and their control mice (WT) at day 3 after 2/3 PH (*n* = 3/group). **g, h** qRT-PCR analyses of *α-SMA* and *Ihh* genes expression in primary HSCs by CA treatment (*n* = 4/group). CA treated primary HSCs which were transfected with Tgr5 siRNA or NCi. **i** qRT-PCR analyses of *Tgr5* gene expression in primary hepatic stellate cells. **j, k** qRT-PCR analyses of *α-SMA* and *Ihh* genes expression in primary HSCs (*n* = 4/group). Error bars in all experiments represent SEM; Significance was determined by unpaired two-tailed Student's *t* test and by one-way ANOVA. *\*P* < 0.05, *\*\*P* < 0.01.

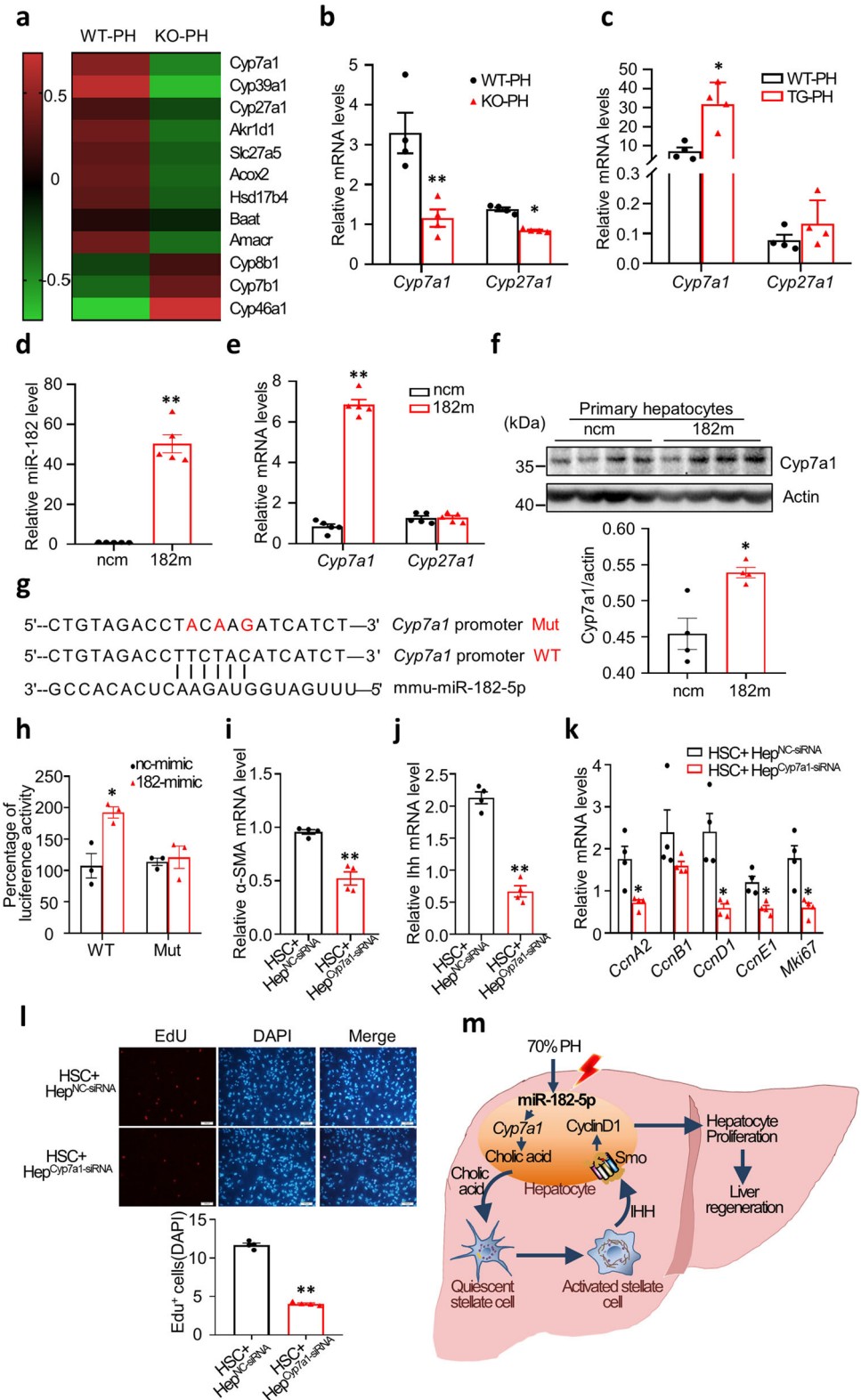

activate Hedgehog signaling in hepatocytes, resulting in hepatocyte proliferation (Fig. 7m). Our study reveals a mechanism underlying the crosstalk between hepatocytes and stellate cells in the liver, which is essential for LR in mice. Our study provides new evidence on the critical role of hepatocyte Hh in LR[27,28] and uncovers a molecular mechanism by which PH activates Hh signaling in LR.

LR is an extremely complex process that involves not only hepatocytes but also other cell types in the liver such as Kupffer cells (liver resident macrophages), stellate cells, and cholangiocytes[4,7–9,35]. The expression levels of miR-182-5p are low in the liver of normal adult mice[36], but are greatly induced by PH in mouse hepatocytes, suggesting a potential role of this molecule in PH-induced LR. Consistent with this view, hepatocyte-specific overexpression of miR-182-5p greatly enhances LR in mice. There is some evidence showing that cholangiocytes and hepatocytes share a common precursor cell in

**Fig. 7 Hepatic miR-182-5p targets Cyp7a1 to promote hepatocyte proliferation. a** The heatmap of DEGs in BA synthesis signaling pathway. Green represents downregulation while red represents up regulation. qRT-PCR analyses of *Cyp7a1* and *Cyp27a1* gene expression in the liver of miR-182-5p KO (**b**) or TG mice (**c**) compared with their respective control mice (3d after PH; *n* = 4/group). miR-182-5p mimic (182 m) or its negative control (ncm) were overexpressed in primary hepatocytes from C57BL/6 J mice. **d, e** miR-182-5p level, and the *Cyp7a1* and *Cyp27a1* genes expression levels were determined by qRT-PCR (*n* = 5/group). **f** The Cyp7a1 protein expression level was determined by western blot. **g** Alignment of the sequences of Cyp7a1 gene promoter and miR-182-5p. **h** Luciferase reporter assay to examine the interactions between miR-182-5p and the predicted target site in the Cyp7a1 gene promoter. Plasmids with the Cyp7a1 gene promoter or mutated gene promoter were co-transfected with miR-182-5p mimic or control mimic into primary hepatocytes. Renilla luciferase activity was measured by a Dual-Glo luciferase assay system and normalized to internal control firefly luciferase activity (*n* = 3 biological Replicates). HSCs co-cultured with Cyp7a1 siRNA (Hep$^{Cyp7a1-siRNA}$) or their control siRNA-treated (Hep$^{NC-siRNA}$) hepatocytes isolated from TG mice. **i, j** qRT-PCR analyses of *α-SMA* and *Ihh* genes expression in HSCs. **k** qRT-PCR analyses of Cyclin genes expression in primary hepatocytes. **l** EdU immunostaining of primary hepatocytes. **m** A proposed model on the mechanism by which miR-182-5p regulates liver regeneration induced by PH. Error bars in all experiments represent SEM; Significance was determined by unpaired two-tailed Student's *t* test and by one-way ANOVA. *$P < 0.05$, **$P < 0.01$.

development[4], and cholangiocytes also play a role in hepatic regeneration during liver injury when hepatocyte proliferation is inhibited[37,38]. However, we found that miR-182-5p levels are greatly induced in mouse hepatocytes isolated from WT mice but not in NPCs isolated from WT mice after PH (Supplementary Fig. 1b). In addition, overexpression of miR-182-5p in primary hepatocytes co-cultured with primary HSCs promotes mouse hepatocyte proliferation (Fig. 4c, d). These results suggest that miR-182-5p in hepatocytes rather than cholangiocytes may play a major role in LR.

Our study demonstrates a BA-mediated paracrine mechanism by which miR-182-5p promotes LR. BAs, which are rapidly increased in mouse plasma and liver after PH[39], have been suggested to play an important role in LR[40–43]. However, how BA biosynthesis is regulated in PH and how this molecule is involved in LR are unknown. We found that miR-182-5p, which is upregulated by PH, stimulates the expression of Cyp7a1, a rate-limiting enzyme in the canonical BA synthesis pathway[31]. Consistent with this finding, CA, a primary BA, was significantly increased in the liver of the miR-182-5p$^{LTG}$ (TG) mice compared to control mice. These results are consistent with the finding that BAs activate HSC in liver[4].

miRNAs are generally considered negative regulators of gene expression[44]. However, we found that treating primary hepatocytes with miR-182-5p mimic had no significant effect on the mRNA levels of known negative regulators of the *Cyp7a1* gene in hepatocytes. On the other hand, we found that miR-182-5p binds to the promoter of *Cyp7a1* gene and positively regulates *Cyp7a1* gene. Interestingly, several studies have previously shown that miRNAs could positively regulate gene expression[34,45]. For example, miR-373 induces the expression of genes with complementary promoter sequences[34]. Our finding provides new evidence to support the positive regulatory mode of miRNAs in regulating gene expression.

In conclusion, our results identified hepatic miR-182-5p as a key regulator of hepatocyte proliferation that promotes LR in mice after 2/3 PH. In addition, we uncover a new signaling mechanism underlying the crosstalk between hepatocytes and HSCs that is critical for PH-induced LR. Our study suggests that upregulation of hepatic miR-182-5p may be a valid therapeutic target to improve LR.

## Methods

**Animal**. miR-182-5p heterozygous knockout mice (miR-182-5p$^{+/-}$) were obtained from the Laboratory Animal Resource Bank, National Institute of Biomedical Innovation, Japan (https://animal.nibiohn.go.jp)[36]. We commissioned Shanghai Model Organisms Center, Inc. (Shanghai, China) to generate a miR-182-5p-Tg$^{flox/flox}$ (ROSA26-polyA-GFP-loxp-lox2272-EF1α (reversed)-loxp-lox2272-miR-182-5p-polyA) mouse line. To generate the liver-specific miR-182-5p overexpressing mice (miR-182-5p$^{LTG}$), we crossed female miR-182-5p-Tg$^{flox/flox}$ mice with male Albumin-cre mice to remove the loxp sites so that continuous expression of miR-182-5p is achieved by the reversed EF1α in the liver. C57BL/6 J mice were purchased from

Slac Laboratory Animal Inc. (Shanghai, China). Mice were housed in a temperature-controlled environment with a 12:12 hr light/dark cycle and had access to food and water *ad libitum*. All animal studies were performed under a protocol approved by the Central South University Animal Care and Use Committee.

**Partial hepatectomy**. The 2/3 PH was performed as described previously[3]. Briefly, 8- to 10-week-old male mice were anesthetized with isoflurane, followed by a ventral midline incision. The left lateral left middle and right middle lobe of the liver (70% of the liver) were removed from the anesthetized mice after pedicle ligation with prolene 6/0. After surgery, the mice were placed on a warming pad for recovery until completely conscious. All partial hepatectomies were performed in the early hours of the morning.

**Cell culture and treatment**. Primary mouse hepatocytes and primary hepatic stellate cells were isolated according to the procedures as described previously[46,47]. For proliferation analysis, primary mouse hepatocytes were transfected with the miR-182-5p mimic, inhibitor, or controls (RiboBio Co. Ltd, Guangzhou, Guangdong, China) for 12 h. Proliferation was induced by treating the cells grown in EdU (5-Ethynyl-2'-deoxyuridine)-containing medium with EGF (30 ng/mL). Thirty-six hours after transfection, hepatocytes were fixed with 4% paraformaldehyde (PFA) and the incorporation of EdU into actively proliferating cells was evaluated using a Cell-Light™ EdU Cell Proliferation Detection kit (RiboBio Co. Ltd, Guangzhou, China) according to the manufacturer's instructions. For co-culture studies, mouse primary hepatocytes were co-cultured with NPC, BMDMs, primary hepatic stellate cells (pri-HSCs), or the HSC cell line LX2 for 48 h, respectively. Primary hepatocytes were treated with the CM of primary hepatocytes co-cultured with stellate cells and the effects of CMs on mouse hepatocyte proliferation were determined by western blot, qRT-PCR, and EdU immunostaining.

**Ki67 staining**. Mouse liver tissues were fixed with 4% paraformaldehyde before embedding them into paraffin blocks. Ki67 staining was performed as previously described[22].

**RNA isolation, real-time quantitative (q)RT-PCR, and western blotting**. Total RNAs were extracted using the TRIzol Reagent (Invitrogen, Life Technologies, NY, USA, Cat#15596026) following the manufacturer's instructions. mRNAs were reverse transcribed using the RevertAid First Strand cDNA Synthesis Kit (Thermo Scientific, USA, K1622). Real-time qPCR was carried out on a 7900HT Fast Real-Time PCR System (Applied Biosystem, USA). The sequences for the primer pairs used in this study are listed in Supplementary Table 1. To examine protein expression levels, cells were lysed in RIPA lysis buffer (Beyotime, Shanghai, China, P0013B). Antibodies to cyclins D1 (Cat#2922), ERK-P (Cat#4377), S6 (Cat#2217), S6-P Cat#2215), 4EBP1 (Cat#9452), and 4EBP1-P (Cat#9459) were from Cell Signaling Technologies, and Cyp7a1(Cat# DF2612) were from Affinity Biosciences. The anti-β-Actin antibody (Cat#A3854) was from Sigma.

**Bile acid levels and composition**. Bile acid levels in the liver were measured by using a bile acid kit (Crystal Chem.; #80471). Individual bile acids were quantified by liquid chromatography-mass spectrometry (LC-MS) as described[48,49]. Bile acid separation was achieved by using an Acuity (Waters, Milford, MA) UPLC BEH C18 column (1.7 microns 2.1 × 100 mm) on a Nextera UPLC (Shimadzu, Tokyo, Japan); the temperature of the column and autosampler was 65 °C and 12 °C, respectively. The sample injection volume was 1 μL. The mobile phase consisted of 10% acetonitrile and 10% methanol in water containing 0.1% formic acid (Mobile Phase A) and 10% methanol in acetonitrile 0.1% formic acid (Mobile Phase B) delivered as a gradient: 0–5 min Mobile Phase B was held at 22%; 5–12 min Mobile Phase B was increased linearly to 60%; 12–15 min Mobile Phase B was increased linearly to 80%; and 15–19 min Mobile Phase B was kept constant at 80% at a flow rate of 0.5 ml/min. The mass spectrometer (Q-Trap 5500; Sciex, Framingham, MA)

was operated in negative electrospray mode working in the multiple reaction mode (MRM). Operating parameters were: curtain gas 30 psi; ion spray voltage 4500 V; temperature 550°C; ion source gas 1 60 psi; ion source gas 2 65 psi. Transition MRMs, declustering potential, entrance potentials, and collision cell exit potentials were optimized using the Analyst software (Sciex, Framingham, MA). Dwell times were 25msec.

**RNA Sequencing.** Total RNA was extracted from mouse liver tissue using RNAiso Plus Total RNA extraction reagent (#9109, TAKARA) following the manufacturer's instructions and checked for a RIN number to inspect RNA integrity by Agilent 2100 bioanalyzer (Agilent Technologies, Santa Clara, CA, US) to confirm the insert size and calculate the mole concentration. Qualified total RNA was further purified by RNAClean XP Kit (Cat A63987, Beckman Coulter). Following purification, the mRNA is fragmented into small pieces using divalent cations. The cleaved RNA fragments are copied into first strand cDNA using reverse transcriptase and random primers. This is followed by second strand cDNA synthesis using DNA Polymerase I and RNase H. These cDNA fragments then go through an end repair process, the addition of a single 'A' base, the polyA containing mRNA molecules was purified using poly-T oligo-attached magnetic beads, and then ligation of the adapters. The products are then purified and enriched using PCR to create the final cDNA library (Inc.Kraemer Boulevard Brea, CA, USA) and RNase-Free DNase Set (Cat#79254, QIAGEN, GmBH, Germany). The cluster was generated by cBot with the library diluted to 10 pM and then sequenced on the Illumina HiSeq 2000/2500 (Illumina, USA). The library construction and sequencing were performed at Shanghai Biotechnology Corporation.

**Luciferase reporter assay.** The promoter sequence for the *Cyp7a1* gene was amplified by PCR with specific primers: forward, 5'-TGCTTCTTGTGAAGTC TTGTGCTGT-3'; reverse, 5'-TTTATAGATAAGCAGATTGCTG-3'. The PCR product was cloned into a pmirGLO vector (Promega) between SacI and SalI sites downstream of the Renilla luciferase gene. To generate the mutant construct, three nucleotides in the target site (CTACAAGA) of the miR-182-5p seed region were mutated (CTTCTACA). For luciferase assays, primary mouse hepatocytes were transfected with the *Cyp7a1* luciferase reporter of mutant or blank vector alone with miR-182-5p mimic mixture or mimic controls using Lipofectamine 3000. Luciferase activity in the cell lysates was measured using the Dual-Glo luciferase assay system according to the manufacturer,s instructions (#E2940, Promega). Luciferase activities were calculated as the ratio of firefly to Renilla luminescence and normalized to the average ratio of the blank control.

**Statistics and reproducibility.** Statistical analyses were carried out using Graph-Pad Prism 8.0 Software (www.graphpad.com) or Microsoft Excel (Microsoft). Data analysis involved unpaired two-tailed Student's *t* test for two groups and one-way ANOVA for more than two groups. All results were presented as the mean ± SEM and $p < 0.05$ was considered to be statistically significant. Data are representative of at least three independent experiments.

**Reporting summary.** Further information on research design is available in the Nature Research Reporting Summary linked to this article.

## Data availability
The source data underlying the graphs and images in the figures are provided in Supplementary Data 1 and 2. The unprocessed gel blot images with size markers are provided in Supplementary Fig. 7. The authors declare that the data supporting the findings of this study are available within the article and from the corresponding author on reasonable request. RNA-Seq data generated in this paper can be accessed at the Gene Expression Omnibus (GEO) repository (accession number: GEO: GSE206451).

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

## Acknowledgements

This work was partially supported by grants 2018YFC2000100 and 2019YFA0801900 from the National Key R&D Program of China and grants 81730022, 82170886, 81800758, and 81870601 from the National Nature Science Foundation of China.

## Author contributions

W.M. and T.X.: collection and assembly of data, and preparation of the first draft of the manuscript; Z.J., J. Wang., J.D., J. Wen., and B.L.: data collection; J.B. and M.L.: supervision of students and contribute to data analysis and discussion; F.L. and W.M.: conceptualization and design, data analysis and interpretation, manuscript writing, financial support, and final approval of the manuscript. All authors reviewed and approved the manuscript. F.L. and W.M. are the guarantors of this work and, as such, had full access to all the data in the study and take responsibility for the integrity of the data and the accuracy of the data.

## Competing interests

All authors declare no potential conflicts of interest related to this work.
