## [Peer Review File · Communications Biology]

Reviewers' comments:

Reviewer #1 (Remarks to the Author):

In this report, Xiao et al. investigated a role of hepatic miR-182-5p in regulating liver regeneration (LR). The two-thirds partial hepatectomy (PH) strongly induced miR-182-5p expression. Suppressing miR-182-5p abrogated PH-induced LR, while liver-specific overexpression of miR-182-5p promotes LR in mice. Moreover, they found a new mechanism by which miR-182-5p increases cholic acid production in hepatocytes to indirectly promote hedgehog ligand production in stellate cells, thereby promoting hepatocyte proliferation.

The experiments are well designed and the results are clear and well presented. The conclusions are of potential interest to contribute to our understanding of cell cross-talk during liver regeneration.

Major comments:

1. During early stages of LR, Cyp7a1 is strongly suppressed. The mechanism by which miR-182-5p increases Cyp7a1 expression and cholic acid production to promote LR needs more justification and discussion. Maybe a time course of miR-182-5p along with Cyp7a1 expression during LR may show the more accurate time point when this mechanism starts to work.
2. Cholic acid(CA) production in hepatocytes, which indirectly promotes hedgehog ligand production in stellate cells is interesting. Is there any report showing how CA activates stellate cells? Any CA receptors (FXR, TGR5, S1PR2, VDR...) in stellate cells may mediate CA's effects on increasing hedgehog ligand production?

Reviewer #2 (Remarks to the Author):

promotes or inhibit liver regeneration in mice with PH. Interestingly, in an in vitro co-culture model they found that hepatic stellate cells (HSC) can stimulate hepatocytes proliferation likely via the activation of hedgehog signaling, which is mediated by bile acids (mainly cholic acid) induced hedgehog production in HSC.

Overall, the findings that HSC may promote hepatocyte proliferation by secretion hedgehog ligand that is stimulated by bile acid are interesting and may have some impact for the liver regeneration field. Nonetheless the manuscript suffered some weaknesses mainly on the lacking of data robustness and relatively descriptive nature of this study. Most experiments were only performed in one time point (DAY 3 after PH) and how these findings would be relevant to the dynamic regenerative course of liver after PH are unclear. There are no experimental data to support/confirm bile acids and cyp7a1 are indeed critical in promoting liver regeneration after PH after manipulating miR-182-5p.

Major:

1. Liver regeneration is a dynamic process. A detail time course data on cell proliferation and biochemical and molecular changes were missing, which is a major weakness for this study. For instance, the proliferation marker, biochemical and molecular signaling changes should be determined on day 1, 2 and 3 (or even day 7 as the authors have collected) after PH such as in supplemental figure 1b.
2. How does miR-182-5p affect MTOR and ERK?
3. Some experimental details were missing. For instance, how long the primary hepatocytes were cultured in these experiments using conditional medium and hedgehog inhibitor?
4. Bile acids uptake require specific transporters such as NTCP. It was unclear whether HSC expression NTCP? In figure 6g-h, what HSCs were used? LX2? If immortalized cells were used, it

was unclear whether this could be true for primary HSC or in vivo?

5. Some key data/information was missing to support the major conclusion of this study. If the hepatocyte proliferation is indeed mediated by bile acids (cholic acid), authors need to use pharmacological approach to sequester bile acids and genetic approach to knockdown cyp7a1 or cross the miR-182 mice with Cyp7A KO mice.

Dear Reviewers:

We were very glad to learn that the reviewers considered our work “well designed” and “the results are clear and well presented”. We also thank you for your constructive comments and suggestions. In accordance with these suggestions, we have carried out additional experiments, addressed all questions, and carefully revised our manuscript (The changes are marked in brown). My point-to-point responses to your comments are shown below and we hope that these revisions meet with your approval.

Thanks.

Feng Liu, Ph.D.

Reviewer #1

1. During early stages of LR, Cyp7a1 is strongly suppressed. The mechanism by which miR-182-5p increases Cyp7a1 expression and cholic acid production to promote LR needs more justification and discussion. Maybe a time course of miR-182-5p along with Cyp7a1 expression during LR may show the more accurate time point when this mechanism starts to work.

Response: We thank the reviewer for this constructive suggestion. As suggested by the reviewer, we performed a time-course study and examined the expression levels of miR-182-5p and Cyp7a1 at various time points during LR. We found that Cyp7a1 expression was low at the early stages but increased at the proliferative stage during LR, concurrently with increased miR-182-5p expression at the later stage (Supplementary Fig. 6b). Importantly, heterozygous knockout of miR-182-5p in mice had no significant effect on Cyp7a1 expression at the early stages of LR but inhibited Cyp7a1 expression at the proliferative phase (Supplementary Fig. 6c). These results suggest a phase-specific role of miR-182-5p in regulating Cyp7a1 expression during LR. We have included these new data in the revised manuscript (page 9, lines 204-209).

2. Cholic acid(CA) production in hepatocytes, which indirectly promotes hedgehog

ligand production in stellate cells is interesting. Is there any report showing how CA activates stellate cells? Any CA receptors (FXR, TGR5, S1PR2, VDR...) in stellate cells may mediate CA's effects on increasing hedgehog ligand production?

Response: It has been reported by Xie et al that CA activates stellate cells via TGR5 signaling pathway (Xie et al., *EBioMedicine*, 2021, 66:103290). In addition, there are some reports showing that the expression levels of FXR, S1PR2 and VDR were low in stellate cells compared to hepatocytes (Peter Fickert et al., *Am J Pathol* 2009, 175:2392–2405; Yoshimitsu Kiriyama et al., *Biomolecules*, 2019, 9:232). We confirmed these findings and found that the expression of these CA receptors was not stimulated by CA treatment (Supplementary Fig. 5k and Supplementary Fig. 5l). However, CA treatment greatly induced the expression of TGR5, a major receptor in stellate cells which mediates BA-induced HSC activation (Supplementary Fig. 5l). We demonstrate that suppressing TGR5 expression by siRNA (Fig. 6i) greatly reduced the expression levels of α -SMA (Fig. 6j) and *Ihh* (Fig. 6k) in CA-treated stellate cells, confirming an important role of TGR5 in CA-induced hedgehog ligand production. We have added these new data in the revised manuscript (Fig. 6i-6k, Supplementary Fig. 5k and 5l and page 8-9, line 184-192).

Reviewer #2

1. Liver regeneration is a dynamic process. A detail time course data on cell proliferation and biochemical and molecular changes were missing, which is a major weakness for this study. For instance, the proliferation marker, biochemical and molecular signaling changes should be determined on day 1, 2 and 3 (or even day 7 as the authors have collected) after PH such as in supplemental figure 1b.

Response: We thank the reviewer for this constructive comment. As suggested by the reviewer, we collected liver tissues at the suggested time points after 70% of PHx and examined several proliferation biomarkers from these tissues (Please see the attached Fig. 1A). We found that the mRNA levels of IL-6 and TNF α were rapidly increased at 2h but declined at 6 h in the regenerating liver after 2/3 PHx (Response Fig. 1B). In addition, the mRNA levels of cyclin D1, A2, and B1 were all significantly induced during 36h-168h and returned to basal levels at 336 hours after 2/3 PHx (Response

Fig. 1C), which is consistent with the three phases of liver regeneration after 2/3 PHx in mice (Response Fig. 1A)(Chen XG, *Biol Res* 2014;47:59; Kang LI, *Cells* 2012;1:1261-1292; Li N, *Oncotarget* 2017;8:3628-3639), suggesting that mouse liver regeneration model is successfully established.

2. How does miR-182-5p affect MTOR and ERK?

Response: In Fig. 5f and 5h, we found that a stimulatory effect of miR-182-5p on mTOR and ERK signaling were significantly blocked by either pharmacological inhibition of hedgehog signaling with cyclopamine (5 μ M) for 24h (Fig. 5f), or by siRNA-mediated suppression of the Hh receptor Smo in primary hepatocytes (Supplementary Fig. 4f and Fig. 5h), suggesting that IHH signaling is involved in miR-182-5p-mediated mTOR and ERK signaling. Thus, it is likely that miR-182-5p enhances IHH signaling, subsequently increases mTOR and ERK signaling in hepatocytes. How does IHH signaling increase mTOR and ERK signaling is beyond the scope of the current study, but could be an interest topic of future investigations.

3. Some experimental details were missing. For instance, how long the primary hepatocytes were cultured in these experiments using conditional medium and hedgehog inhibitor?

Response: We thank the reviewer for this constructive comment and have added more information on experimental details in the revised manuscript (page 7, lines 157; page

8, lines 181). In addition, we have added information on the method of RNA Sequencing in the revised manuscript (page 14, lines 332-346).

4. Bile acids uptake require specific transporters such as NTCP. It was unclear whether HSC expression NTCP? In figure 6g-h, what HSCs were used? LX2? If immortalized cells were used, it was unclear whether this could be true for primary HSC or in vivo?

Response: We thank the reviewer for raising this point. However, NTCP expression was very low or even undetectable in stellate cells (*Peter Fickert et al., Am J Pathol 2009, 175:2392–2405*), suggesting that NTCP may not be involved in BA-induced HSC activation. In Fig. 6g-h of our previous manuscript, we used the hepatic stellate cell line LX2. To address the reviewer's question, we have repeated the experiments using primary HSCs. We found that the expression levels of α -SMA (Fig. 6g) and *Ihh* (Fig. 6h) are also increased in primary HSC by CA treatment, which is consistent with the data shown in previous Fig. 6g-h using LX2 cells.

5. Some key data/information was missing to support the major conclusion of this study. If the hepatocyte proliferation is indeed mediated by bile acids (cholic acid), authors need to use pharmacological approach to sequester bile acids and genetic approach to knockdown *cyp7a1* or cross the miR-182 mice with *Cyp7A* KO mice.

Response: We thank reviewer for this constructive suggestion. To address the question, we treated primary hepatocytes isolated from TG mice with *Cyp7a1* siRNA (Supplementary Fig. 6e) and co-cultured these cells with primary HSCs. Suppressing *Cyp7a1* expression greatly reduced the expression levels of α -SMA (Fig. 7i) and *Ihh* (Fig. 7j) in primary HSCs as well as the promoting effect of HSC+Hep^{182TG} on hepatocyte proliferation (Figs. 7k-7l). These results provide further evidence that hepatocyte *Cyp7a1* plays a critical role in miR-182-5p-induced stellate cell hedgehog signaling and hepatocyte proliferation. We have added these new data and information in the revised manuscript (Fig. 7i-7l, Supplementary Fig. 6e and page 10, line 221-228).

REVIEWERS' COMMENTS:

Reviewer #2 (Remarks to the Author):

The authors have sufficiently addressed the previous review comments. It is now acceptable for publication in COMMSBIO.

Reviewer #3 (Remarks to the Author):

Authors have addressed my concerns and I am satisfied for the revision.